# AB186 Inhibits Migration of Triple-Negative Breast Cancer Cells and Interacts with α-Tubulin

**DOI:** 10.3390/ijms23126859

**Published:** 2022-06-20

**Authors:** Marine Geoffroy, Marine Lemesle, Alexandra Kleinclauss, Sabine Mazerbourg, Levy Batista, Muriel Barberi-Heyob, Thierry Bastogne, Wilfrid Boireau, Alain Rouleau, Dorian Dupommier, Michel Boisbrun, Corinne Comoy, Stéphane Flament, Isabelle Grillier-Vuissoz, Sandra Kuntz

**Affiliations:** 1CNRS, CRAN, Université de Lorraine, F-54000 Nancy, France; mgeoffroy@genethon.fr (M.G.); marine.lemesle@univ-lorraine.fr (M.L.); alexandra.kleinclauss@univ-lorraine.fr (A.K.); sabine.mazerbourg@univ-lorraine.fr (S.M.); levy.batista@univ-lorraine.fr (L.B.); muriel.barberi@univ-lorraine.fr (M.B.-H.); thierry.bastogne@univ-lorraine.fr (T.B.); stephane.flament@univ-lorraine.fr (S.F.); isabelle.grillier-vuissoz@univ-lorraine.fr (I.G.-V.); 2FEMTO-ST Institute, CNRS, Université de Bourgogne Franche-Comté, F-25030 Besançon, France; wboireau@femto-st.fr (W.B.); alain.rouleau@femto-st.fr (A.R.); 3CNRS, L2CM, Université de Lorraine, F-54000 Nancy, France; d.dupommier@gmail.com (D.D.); michel.boisbrun@univ-lorraine.fr (M.B.); corinne.comoy@univ-lorraine.fr (C.C.)

**Keywords:** breast cancer, troglitazone derivatives, apoptosis, cytoskeleton, tubulin, cell migration

## Abstract

Breast cancer is one of the leading causes of cancer-related death among females worldwide. A major challenge is to develop innovative therapy in order to treat breast cancer subtypes resistant to current treatment. In the present study, we examined the effects of two Troglitazone derivatives Δ2-TGZ and AB186. Previous studies showed that both compounds induce apoptosis, nevertheless AB186 was a more potent agent. The kinetic of cellular events was investigated by real-time cell analysis system (RTCA) in MCF-7 (hormone dependent) and MDA-MB-231 (triple negative) breast cancer (TNBC) cells, followed by cell morphology analysis by immuno-localization. Both compounds induced a rapid modification of both impedance-based signals and cellular morphology. This process was associated with an inhibition of cell migration measured by wound healing and transwell assays in TNBC MDA-MB-231 and Hs578T cells. In order to identify cytoplasmic targets of AB186, we performed surface plasmon resonance (SPR) and pull-down analyses. Subsequently, 6 cytoskeleton components were identified as potential targets. We further validated α-tubulin as one of the direct targets of AB186. In conclusion, our results suggested that AB186 could be promising to develop novel therapeutic strategies to treat aggressive forms of breast cancer such as TNBC.

## 1. Introduction

Breast cancer is one of the leading causes of cancer-related death among females worldwide [1]. Breast tumors are classified by the presence of three molecular markers: estrogen receptor (ER), progesterone receptor (PR) and human epidermal growth factor receptor 2 (HER2), that allow targeted therapy beside chemotherapy [2]. Nevertheless, the effects of therapeutic agents are limited by de novo or acquired resistance and they often cannot avoid the metastatic spread [3,4]. Triple-negative breast cancer (TNBC) subtype (ERα-, PR-, HER2-) presents the worst prognosis due to its highly invasive nature and relatively low response to therapeutics [5]. Therefore, there is an urgent need to develop alternative therapeutics.

Troglitazone (TGZ) is a synthetic compound of the thiazolidinediones (TZDs) family, such as rosiglitazone (RGZ) and pioglitazone (PGZ), used as antidiabetic agents. TZDs also display anti-tumoral activity not only in vitro but also in vivo [6,7]. Despite promising in vitro data and pre-clinical studies highlighting the strong potential of TZDs in breast cancer treatment, clinical trials did not display clear beneficial effects. Their anticancer properties were renewed by a phase II trial on chronic myeloid leukemia. This study provided evidence that in combination with imatinib, PGZ improved the erosion of the cancer stem cells pool [8,9]. Similarly, a clinical study indicated that efatutazone (another TZD derivative) associated with paclitaxel could be effective in anaplastic thyroid carcinoma [10]. More recently, Ronen et al. showed that RGZ in combination with MEK inhibitors enhanced epithelial differentiation and adipogenesis of breast cancer cells both in vitro and in vivo [11]. Unfortunately, the majority of TZDs were withdrawn from the market because of their side effects, especially severe hepatic toxicity for TGZ [12]. In this context, our team developed new TGZ derivatives that are more potent and less toxic for the liver. We synthetized the already known ∆2-troglitazone (∆2-TGZ) and also the original molecule AB186 (Figure 1) [13,14,15]. Both derivatives possess a double bond adjoining the terminal thiazolidine-2,4-dione ring of TGZ abrogating the PPAR γ agonist activity [16,17,18,19]. In addition to ∆2-TGZ, AB186 is deoxygenated on the chromane moiety and holds at this position an aminoalkyl chain, both inducing higher efficiency to inhibit cell proliferation of breast cancer cells MCF-7 and MDA-MB-231 [14].

Previously, we have demonstrated that compared to TGZ, ∆2-TGZ and AB186 exhibit a lower toxicity toward human non-malignant hepatocytes in primary culture and an improved anti-proliferative activity on breast cancer cell lines [13,14]. We showed that ∆2-TGZ induced apoptosis in luminal cells MCF-7 and in MDA-MB-231 and Hs578T TNBC cells [20,21]. In the present study, we aimed to better understand the mechanism of action of Δ2-TGZ and AB186. We observed a stronger pro-apoptotic effect of AB186 compared to ∆2-TGZ in breast cancer cells and interestingly no effect in the non-tumoral MCF-10A cells after 24 h of treatment. Then, we analyzed the dynamic response to ∆2-TGZ and AB186 using a real-time cell analysis on MCF-7 and MDA-MB-231 cells. Surprisingly, we observed an early modification of impedance-based signal correlated with cell morphology modifications between 2 and 4 h. These events were followed by cell migration inhibition in MDA-MB-231 and Hs578T TNBC cells at 6 h. These data suggested interactions with cytoskeleton proteins. In order to identify the targets of AB186, we performed surface plasmon resonance (SPR) analysis that comforted this hypothesis. We identified six targets from the cytoskeleton: γ-actin, β-tubulin, filamin-A, α-tubulin, moesin, talin. Finally, pull-down assays in MDA-MB-231 and Hs578T and in vitro tubulin polymerization assays highlighted the binding of AB186 to α-tubulin.

## 2. Results

### 2.1. AB186 Is a More Potent Inducer of Apoptosis Compared to ∆2-TGZ in MDA-MB-231 Breast Cancer Cells and Is Ineffective in the Non-Malignant MCF-10A

AB186 is a new TGZ derivative which possesses, in addition to the double bond adjoining the thiazolidine-2,4-dione ring as in ∆2-TGZ, a deoxygenation of the chromane heterocycle and a hydrophobic aminoalkyl moiety (Figure 1).

In a previous study, we demonstrated that Δ2-TGZ induced apoptosis after 24 h of treatment in MDA-MB-231 breast cancer cells but not in non-tumorigenic mammary cells MCF-10A [21]. We exposed these cells to 4 µM of AB186, a dose that was just above the cell viability IC50 that we previously measured in MDA-MB-231 [14]. AB186 also led significantly to the apoptosis of MDA-MB-231 breast cancer cells but not of MCF-10A cells (Figure 2). After 24 h of exposure, 4 µM of AB186 induced 44% of apoptotic cells while 20 µM of Δ2-TGZ was required to obtain 32% of apoptotic cells. Likewise, after 48 h of treatment, Δ2-TGZ and AB186 induced 51% and 55% of apoptotic cells respectively. Thus, AB186 (4 µM) induced a higher or similar apoptosis rate at a five-fold lower dose compared to Δ2-TGZ (20 µM) in these breast cancer cells.

### 2.2. Variations of Impedance Occur Early after ∆2-TGZ and AB186 Treatment in MCF-7 and MDA-MB-231 Breast Cancer Cells

In order to investigate the kinetics of events leading to the anticancer effect of Δ2-TGZ and AB186, we performed a real-time cell analysis (RTCA) allowing the measurement of impedance-based signals. MCF-7 and MDA-MB-231 cells were exposed to Δ2-TGZ (25 µM and 20 µM respectively) or AB186 (5 µM and 4 µM respectively). In both control cells, the Cell Index (CI) value increased progressively until the end of the experiment to reach 2.4 and 1.9 in MCF-7 and MDA-MB-231 respectively (Figure 3). In both cell lines, kinetics showed a rapid and transient increase of CI after Δ2-TGZ and AB186 exposure compared to control cells. Finally, after 24 h, the CI decreased to reach a value equal or below those measured at the beginning of the treatment.

In order to better correlate the impedance modifications to a biological effect, we determined the time of maximal difference of CI between control and treated cells (Table 1). In MCF-7 cells, this key time point was observed at 1 h 55 min ± 8 min and 4 h ± 75 min in response to Δ2-TGZ and AB186 respectively (Figure 3A, Table 1). In MDA-MB-231 cells, the CI reached a maximal difference at 1 h 50 min ± 5 min and 3 h 55 min ± 13 min after Δ2-TGZ and AB186 exposure respectively (Table 1).

These data showed that early modifications induced by both compounds occurred long before the apoptotic process, and could be a signature of cell morphology modification.

### 2.3. ∆2-TGZ and AB186 Induce Early Morphological Changes in MCF-7 and MDA-MB-231 Breast Cancer Cells

In order to test this hypothesis, we analyzed by immunofluorescence the morphology of MCF-7 and MDA-MB-231 cells using three cytoskeletal components: actin, α-tubulin, and vimentin after 2 h 30 min, 4 h 30 min, 8 h 30 min, or 10 h 30 min of treatment. The control MCF-7 cells displayed a cobblestone-like phenotype with strong cell–cell adhesion (Figure 4A). In response to Δ2-TGZ and AB186, MCF-7 cell morphology was altered by changes in cell shape and size (Figure 4B,C). We observed some elongated cells around 4 h 30 min of exposure to Δ2-TGZ and AB186 and some shrunken cells with condensed chromatin after 8 h 30 min of treatment. At 10 h 30 min of treatment, almost all cells were shrunken.

Control MDA-MB-231 cells displayed a fibroblast-like morphology with a pronounced cellular scattering (Figure 5A). In response to Δ2-TGZ and AB186, MDA-MB-231 cell morphology was also altered with elongated cells harboring an unorganized actin network from 2 h 30 min of treatment (Figure 5B,C). As observed in MCF-7 cells, both treatments led to shrunken cells from 8 h 30 min exposure in MDA-MB-231 cells.

Altogether these data confirmed early alterations of the cell morphology, before 4 h 30 min, in agreement with changes in the impedance-based cell signal.

### 2.4. ∆2-TGZ and AB186 Inhibit Migration Properties of Two Highly Metastatic TNBC Cell Lines: MDA-MB-231 and Hs578T

The modifications of cell shape and the disruption of cytoskeleton induced by Δ2-TGZ and AB186 suggested that these compounds could modify the migration potency of the cells. Indeed, it is well described that cytoskeleton modifications can influence cell migration properties [22]. To determine whether Δ2-TGZ and AB186 could affect cell migration, we performed migration assays on MDA-MB-231 and Hs578T, two TNBC cell lines with highly metastatic properties. First, both cells were exposed to TGZ derivatives and wound healing assay was performed after 3, 6, and 12 h of treatment. We observed a significant wound healing inhibition at 6 h of exposure to Δ2-TGZ and AB186 in MDA-MB-231 cells (Figure 6A). In Hs578T cells, we also observed an inhibition of wound healing that was significant at 6 h and 12 h of Δ2-TGZ and AB186 treatment respectively. Then, we confirmed these results using the transwell assay after 9 h of exposure to the TGZ derivatives. In MDA-MB-231 cells, the number of migrated cells exhibited a significant decrease of 27.16% and 34.70% after Δ2-TGZ and AB186 treatment, respectively, compared to control cells (Figure 6B). In the same way, Δ2-TGZ and AB186 reduced significantly the migration by 41.11% and 49.33% respectively in Hs578T cells compared to control cells.

### 2.5. AB186 Acts through Its Binding to α-Tubulin

Early morphological changes mediated by TGZ derivatives could be explained by the interaction between the drug and the cytoskeleton. In order to determine the localization of AB186 in the cell, we exposed MDA-MB-231 cells to rhodamine-labelled AB186 (ABRhod) for 1 or 6 h (Figure 7). No staining was observed in the nucleus, the signal was diffused after 1 h and concentrated in the cytoplasm after 6 h. These results suggest a cytoplasmic target for AB 186.

To identify direct cytoplasmic targets of AB186, a SPR binding analysis was performed using MDA-MB-231 cytoplasmic extracts and AB186 immobilized on sensor chips [23]. On-Chip mass spectrometry analysis revealed that AB186 could bind to 94 proteins including 6 cytoskeletal proteins, belonging to the similar GO pathways “structural molecule activity” (GO.0005198) and “structural constituent of cytoskeleton” (GO.0005200): γ-actin, β-tubulin, filamin-A, α-tubulin, moesin and talin (Table 2).

By pull-down assay, we validated that biotinylated AB186 (ABBiot) binds to α-tubulin in MDA-MB-231 and Hs578T cells while the interaction between AB186 and β-actin was observed only in MDA-MB-231 cells (Figure 8A). No clear interaction was confirmed with moesin and talin. In addition, AB186 promotes tubulin polymerization into microtubules in vitro. Indeed, tubulin polymerization started earlier compared to the control condition (Figure 8B). AB186 displayed a different profile compared to paclitaxel, well-known to promote microtubule assembly and affect microtubule dynamics. In the presence of AB186, the accumulation of polymerized tubulin started later than in the presence of paclitaxel. The morphology of microtubules was then examined in breast cancer cells by immunofluorescence microscopy (Figure 8C). At room temperature (RT), AB186 induced a dose-dependent enlargement of tubulin network, with elongated microtubules. By contrast, after paclitaxel treatment microtubules were more condensed and reorganized into thick ring-like bundles near the nucleus. Experiments were also performed at 4 °C since cold treatment is known to induce depolymerization of microtubules [24]. As expected, in control cells, the microtubule network was altered by cold treatment. In the presence of paclitaxel, control treatment, no network remodeling was observed between RT and 4 °C conditions. Interestingly, AB186 treatment induced a dose-dependent stabilization of the network compared to control cells, suggesting an active role on the microtubule through binding to α-tubulin (Figure 8C).

## 3. Discussion

The chemo-resistance of TNBC tumors highlights the urgent need to develop new drugs in order to stretch the range of chemotherapy drugs to fight the acquired resistances. Here, we studied two TGZ derivatives, Δ2-TGZ and AB186, that we have previously shown to inhibit cell growth and to induce breast cancer cell death [14,20]. Interestingly, AB186 is a more potent inhibitor of TNBC cell proliferation than efatutazone, a TZD used in combination with chemotherapy drug to treat cancer [14,25]. Our aim was to better understand the mechanism of action of these two derivatives.

In order to investigate the kinetic of events leading to anti-cancerous effect of Δ2-TGZ and AB186, we measured cell impedance in real time. The profiles obtained from RTCA suggested a rapid modification of the cell morphology occurring between 2 and 4 h of treatment with these compounds in both MCF-7 and MDA-MB-231 cells. We then showed an inhibition of MDA-MB-231 and Hs578T cell migration as soon as 6 h of treatment by Δ2-TGZ or AB186. These results were consistent with the fact that TZDs have been reported to inhibit cancer cell motility in vitro and in vivo [26]. Several studies have shown that the inhibition of cell migration by TGZ was associated with the modification of cell morphology [27,28,29]. PK-1 (pancreas), ES-2 (ovary), and MDA-MB-231 (breast) cancer cells were smaller and their shape was modified to a spindle-shaped morphology. In PK-1 cells, the morphological changes induced by TGZ required a remodeling of actin network cells where TGZ inhibited lamellipodia formation and actin polymerization [27,28]. Likewise, in ES-2 cells, TGZ reduced the number of focal adhesions related to actin stress fibers decrease [29]. Moreover, TGZ reduced cell adhesion and spreading of MDA-MB-231 after 1 h of treatment on fibronectin coated-plates [28]. In our study, we also observed changes in cell shape and size. At early time, cells stretched and formed cytoplasmic extensions, followed by a round shape after 10 h 30 of treatment suggesting cell death. These changes were associated with reorganization of tubulin and actin networks.

Using proteomic analyses, we identified six potential cytoskeleton targets of AB186: γ-actin, β-tubulin, filamin-A, α-tubulin, moesin, and talin. Pull-down assays performed on MDA-MB-231 and Hs578T confirmed the binding of AB186 to α-tubulin. In vitro polymerization assays suggested that AB186 stimulated tubulin polymerization into microtubules. Immunofluorescence analysis showed a rapid enlargement and elongation of tubulin network, in agreement with the early maximal difference of CI measured by impedance analysis. This enlargement could result in stronger adhesion properties as previously observed [30]. It remains unclear at present how AB186 exerts its effect on microtubule assembly and stability. It could act as a tubulin stabilizer by a different process than that of paclitaxel. One may suggest that it could work similarly to Mdp3, a microtubule-binding protein that regulates microtubule assembly and stability [31]. Our results corroborate a previous study suggesting that TZDs could act as cytoskeleton binding molecules [32]. The exact consequences of AB186 interaction with α-tubulin need to be investigated.

In conclusion, we determined that AB186 was a potent pro-apoptotic agent in breast cancer cells. We further characterized early modifications of cell morphology followed by the inhibition of cell migration after Δ2-TGZ and AB186 treatment. We identified for the first-time cytoskeleton targets, including α-tubulin, for AB186 that could explain the early morphological changes and migration inhibition and maybe apoptosis. Indeed, if microtubule alterations affect mitosis duration, this could lead to apoptosis [33]. Overall, these data suggest that AB186 could be a promising TZD molecule for TNBC treatment regarding its anti-migratory and pro-apoptotic properties. In order to comfort our hypothesis, this study needs to be extended in vivo by xenograft experiments on mice model.

## 4. Materials and Methods

### 4.1. Materials

The synthesis of ∆2-TGZ and AB186 were performed as previously described [13,14]. In stock solutions, both compounds were dissolved at 50 mM in DMSO. The synthesis of biotinylated and rhodamine-linked AB186 compounds are described in Appendix A). The human breast cancer and the non-tumorigenic human breast epithelial cell lines were purchased from the American Type Culture Collection (Molsheim, Strasbourg). Dulbecco’s modified Eagle medium (DMEM), DMEM/F12, RPMI Medium 1640 (RPMI), Trypsin-EDTA, and PBS were purchased from Life Technologies (Saint Aubin, France). Ethanol (EtOH), dimethylsulfoxide (DMSO), fetal calf serum (FCS), horse serum, human epidermal growth factor (hEGF), L-glutamine, bovine insulin, hydrocortisone and cholera toxin were purchased from Sigma-Aldrich (St. Quentin Fallavier, France).

### 4.2. Methods

#### 4.2.1. Cell Culture and Treatment

MCF-7 and MDA-MB-231 cell lines were cultured at 37 °C under 5% CO_2_ in phenol red DMEM and RPMI respectively, supplemented with 10% FCS and 2 mM L-glutamine. Hs578T cells were grown in DMEM containing phenol red supplemented with 10% FCS, 2 mM L-glutamine, and 10 µg/mL bovine insulin. MCF-10A cells were grown in DMEM/F12 supplemented with 5% horse serum, 2 mM L-glutamine, 100 U/mL penicillin, 100 µg/mL streptomycin, 10 µg/mL bovine insulin, 0.5 µg/mL hydrocortisone, 100 ng/mL cholera toxin, and 20 ng/mL hEGF. Cells were treated with 0.01% DMSO (vehicle) or various concentrations of Δ2-TGZ and AB186 in 1% FCS containing medium.

#### 4.2.2. Flow Cytometry

MDA-MB-231 (4 × 10^5^ cells/well) and MCF-10A (1.5 × 10^5^ cells/well) cells were seeded in 6-well plates and treated with DMSO (vehicle), 20 µM of Δ2-TGZ, or 4 µM of AB186 for 24 h or 48 h. Then, flow cytometry analyses were performed as described previously [21].

#### 4.2.3. Real-Time Impedance-Based Cell Analysis

MCF-7 (8 × 10^3^ cells/well) and MDA-MB-231 (4 × 10^3^ cells/well) cells were seeded into 96-wells microelectronic standard E-plate and grown for 29 h before exposure to DMSO, Δ2-TGZ, or AB186. MCF-7 and MDA-MB-231 cells attachment, proliferation, and size variations were monitored in real-time and measured as impedance using the xCELLigence system (Real Time Cell Analyzer Single Plate (RTCA SPH) system) (Roche Applied Science, Mannheim, Germany) [34]. Cell index (CI) is an arbitrary unit representing the result of the impedance induced by adherent cells to the electron flow. CI was calculated as follows: CI = (impedance at time point n-impedance in the absence of cells)/nominal impedance value. Background impedance of the E-Plate was first determined before seeding the cells by the addition of 50 µL culture medium to each well and subtracted automatically by the RTCA software following the equation: CI = (Zi − Z0)/15 with Zi as the impedance at any given time point and Z0 as the background signal. Impedance measurements were performed every 15 min.

#### 4.2.4. Immunofluorescence

MCF-7 (4 × 10^4^ cells/well) and MDA-MB-231 (8 × 10^4^ cells/well) were seeded in 24-well plates on coverslips and treated with Δ2-TGZ or AB186 for different times. Then, immunofluorescence experiments were performed as described previously [21]. Polymeric F-actin filaments were labelled with Phalloïdin conjugated to Alexa Fluor 488 (A12379, Thermofisher Scientific, Illkirch-Graffenstaden, France) diluted to 1:30. Microtubules and intermediate filaments were labelled with α-tubulin (GTX102079, GeneTex, Letchworth Garden City, England) and vimentin (MAB1633, Chemicon, Nürnberg, Germany) antibodies, respectively, diluted to 1:250. MDA-MB-231 (3 × 10^5^ cells/well) were also seeded in 12-well plates on coverslips and exposed to 3 µM of rhodamine-labelled AB186 (ABRhod) or unlabeled compound (Control) during 1 and 6 h. Then, cells were fixed with 4% of paraformaldehyde and nuclei were labelled with Hoechst. Fluorescence labeling was observed under an Eclipse 80i microscope and images were collected using NIS-Elements Basic Research software, 4.20.00 (Nikon, Champigny sur Marne, France).

#### 4.2.5. Wound Healing Assay

MDA-MB-231 (6 × 10^5^ cells/well) and Hs578T (2 × 10^5^ cells/well) were seeded in 6-well plates for 24 h until 90% of confluence. A scratch was made with a 200 µL pipette tip through the cell monolayer and cells were treated with Δ2-TGZ or AB186 for 0, 3, 6, or 12 h. Wound closure was observed as previously described [35].

#### 4.2.6. Migration Assay

MDA-MB-231 (13,750 cells/well) and Hs578T (13,750 cells/well) were seeded in the upper chamber of Transwell membranes (6.5 mm Transwell*^®^* 8.0 µm pore Polyester Membrane Insert, #3464, Corning) and treated with Δ2-TGZ or AB186 in medium supplemented with 1% FCS. The upper chamber of insert containing cells was placed into the lower chamber supplemented with 10% FCS as chemoattractant. After 9 h of treatment, the migrating cells were analyzed as described previously [35].

#### 4.2.7. SPR Binding Analysis

Cytoplasmic proteins of MDA-MB-231 cells were extracted using the ProteoExtract*^®^* Subcellular Proteome Extraction Kit following manufacturer instructions (Merk Millipore). The cytoplasmic fraction was filtrated through a 0.2 µm microspin Costar at 10,000× *g* for 7 min at 4 °C. Proteins were diluted to 5 µg/mL with PBS 1X before Surface Plasmon Resonance analysis on AB186 hybridized chips (Appendix A).

#### 4.2.8. Pull down Assay

Pull-down assay was performed with Dynabeads M-280 Streptavidin (Life Technologies, Saint Aubin, France) according to the manufacturer’s protocol. Briefly, 50 mM of biotinylated AB186 were fixed to the microbeads during 20 min at room temperature under agitation. Then, the microbeads were washed three times with 0.1% Triton X-100 in PBS buffer and 400 µg of proteins was added for 40 min at 4 °C under agitation. For the control, proteins were incubated with the microbeads without biotinylated AB186. After washing, proteins were re-suspended in lysis buffer while the beads were removed with a magnet and the protein was subjected to Western blot as described previously [21]. The antibodies raised against β-actin (SC-1615, Santa Cruz, Heidelberg, Germany) and α-tubulin (GTX102079, GeneTex, Letchworth Garden City, England) were diluted at 1:5000 and 1:4000 respectively.

#### 4.2.9. Tubulin Polymerization Assay

The effect of AB186 on tubulin assembly was measured using the In Vitro Tubulin Polymerization Assay Kit according to the manufacturer’s instructions (Millipore, Molsheim, France). Briefly, polymerization reactions occur in 70 µL final volume of 1× polymerization buffer (PB)-GTP (1mM final), of which 60 µL is the 60 µM tubulin and 10 µL is the test substance AB186 (5 µM) or paclitaxel (PTX) (10 μM) or DMSO (control). Tubulin assembly was measured by spectrophotometry every 30 s at 350 nm for 90 min at 37 °C.

#### 4.2.10. Statistical Analysis

The results of each experiment are expressed as mean ± standard error of the mean (SEM) of three to six different experiments. Statistical differences were determined using Student’s *t*-test or ANOVA test with Bonferroni post-hoc. Differences in which *p* was less than 0.05 were statistically significant (SPSS v11.0 Computer Software, Chicago, IL, USA).

## Figures and Tables

**Figure 1 ijms-23-06859-f001:**
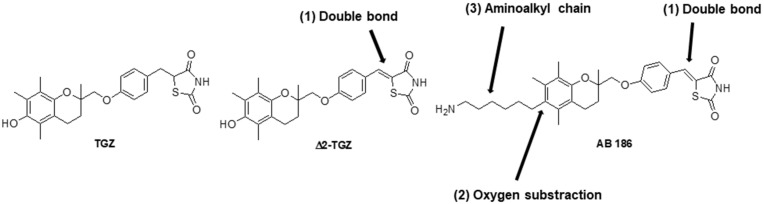
Structures of TGZ, ∆2-TGZ and AB186 compounds. ∆2-TGZ and AB186 are derived from the original TGZ and both of them possess a double bond (1) adjoining the terminal thiazolidine-2, 4-dione ring. AB186 also presents a deoxygenation of the chromane heterocycle (2) and a hydrophobic moiety (3).

**Figure 2 ijms-23-06859-f002:**
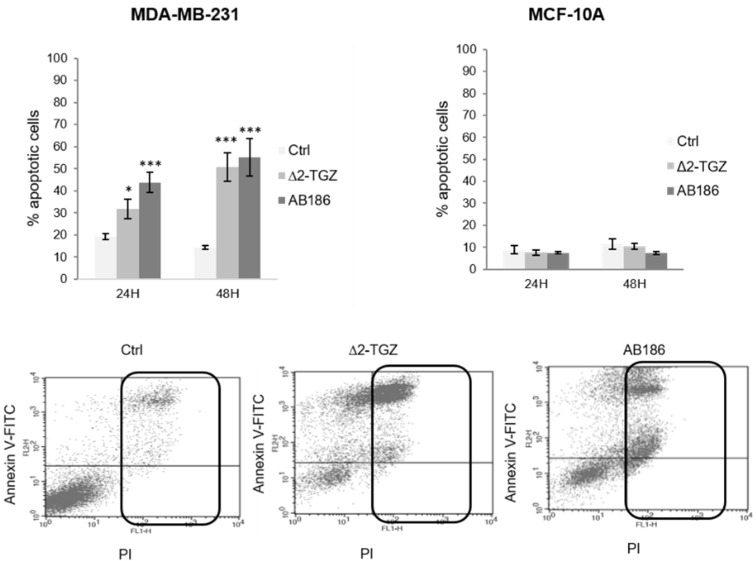
AB186 induces apoptosis in MDA-MB-231 TNBC cells. After treatment for 24 and 48 h with DMSO (Ctrl), 20 µM of Δ2-TGZ or 4 µM of AB186, MDA-MB-231 and MCF-10A cells were co-stained with FITC-Annexin V/PI and analyzed by FACS. The percentage of total apoptotic cells corresponds to Annexin V positive cells and Annexin V/PI positive cells (dot plot of MDA-MB-231 cells exposed to 48 h of treatment). The values represent the means ± SEM of three to six different experiments. ANOVA test with Bonferroni post-hoc was used to determine significant difference from control cells, where * *p* < 0.05 and *** *p* < 0.001.

**Figure 3 ijms-23-06859-f003:**
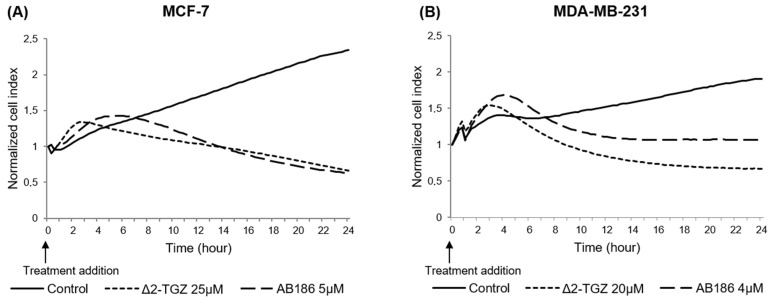
Kinetics of real-time impedance changes in response to TGZ derivatives in MDA-MB-231 and MCF-7 cells. (**A**) MCF-7 and (**B**) MDA-MB-231 cells were grown during 29 h and then were exposed (treatment addition) to Δ2-TGZ (25 µM and 20 µM respectively) or AB186 (5 µM and 4 µM respectively). Control cells were treated with DMSO. Cell index was monitored during 24 h after treatment addition. Reported data are the means of three independent experiments.

**Figure 4 ijms-23-06859-f004:**
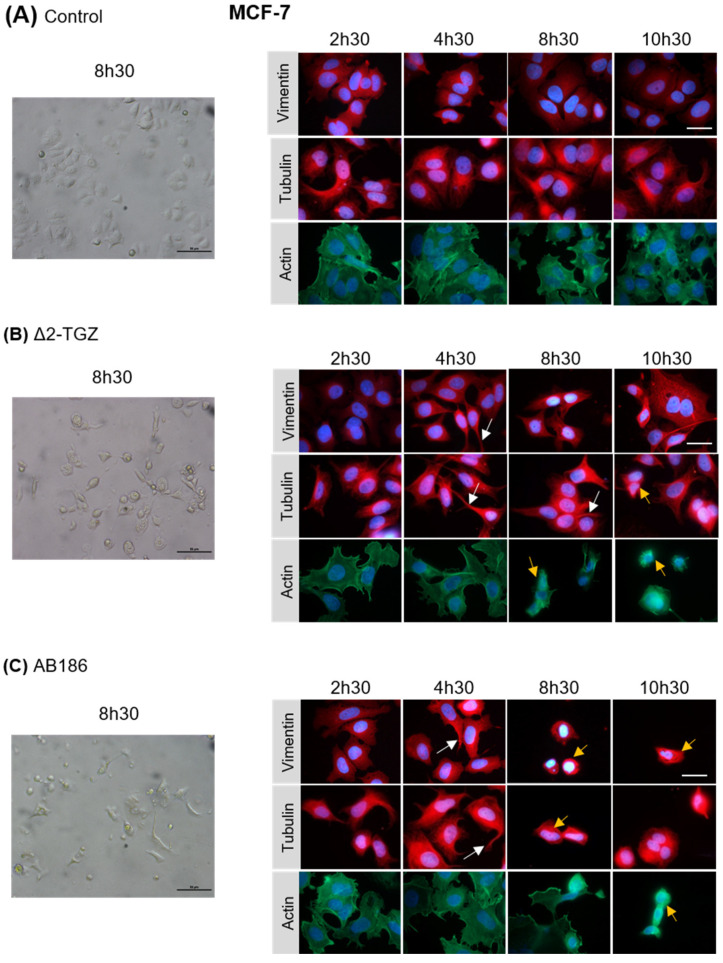
TGZ derivatives alter MCF-7 morphology. MCF-7 cells were treated with DMSO as control (**A**), 25 µM of Δ2-TGZ (**B**) or 5 µM of AB186 (**C**) during 2 h 30 min, 4 h 30 min, 8 h 30 min, and 10 h 30 min. Then, cells were subjected to immunofluorescence analysis and stained with antibodies against vimentin, α-tubulin, and actin. Cell nuclei were labelled with Hoechst dye. Bright field pictures of control and treated cells are also shown after 8 h 30 min. Δ2-TGZ and AB186 led to a disruption of cell–cell contact compared to untreated cells. Elongated cells (white arrows) and some shrunken cells (yellow arrows) were observed. Bar represent 50 µm.

**Figure 5 ijms-23-06859-f005:**
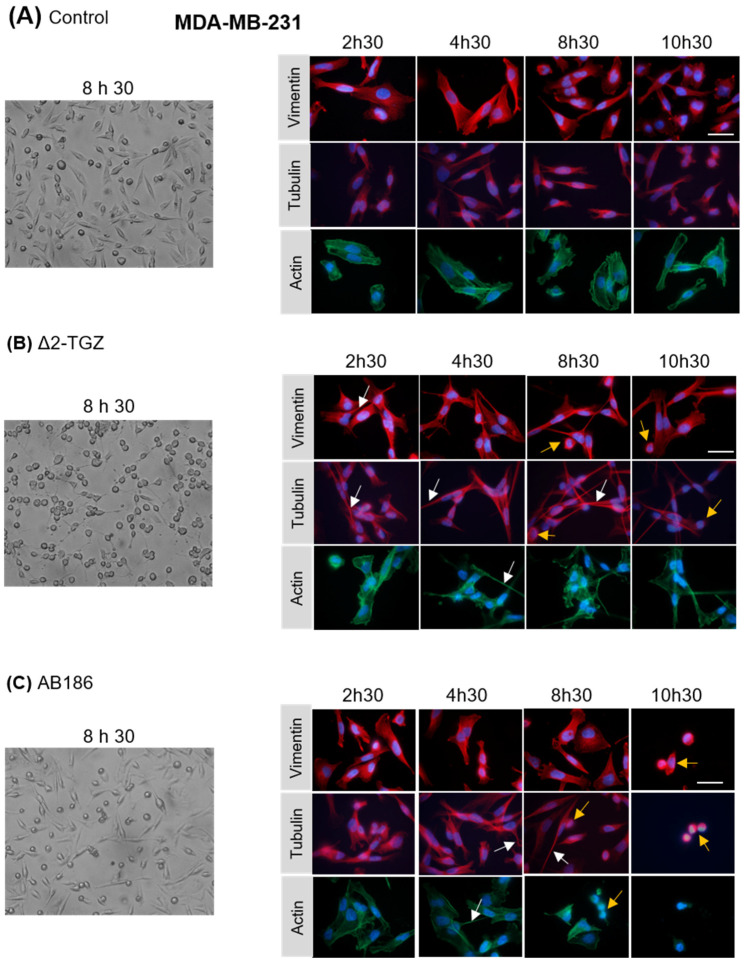
TGZ derivatives alter MDA-MB-231 morphology. MDA-MB-231 cells were treated with DMSO as control (**A**), 20 µM of Δ2-TGZ (**B**) or 4 µM of AB186 (**C**) during 2 h 30 min, 4 h 30 min, 8 h 30 min and 10 h 30 min. Then, cells were subjected to immunofluorescence analysis and stained with antibodies raised against vimentin, α-tubulin and actin. Cell nuclei were labelled with Hoechst dye. Bright field pictures of control and treated cells are also shown after 8 h 30 min. Elongated cells (white arrows) and some shrunken cells (yellow arrows) were observed. Bar represents 50 µm.

**Figure 6 ijms-23-06859-f006:**
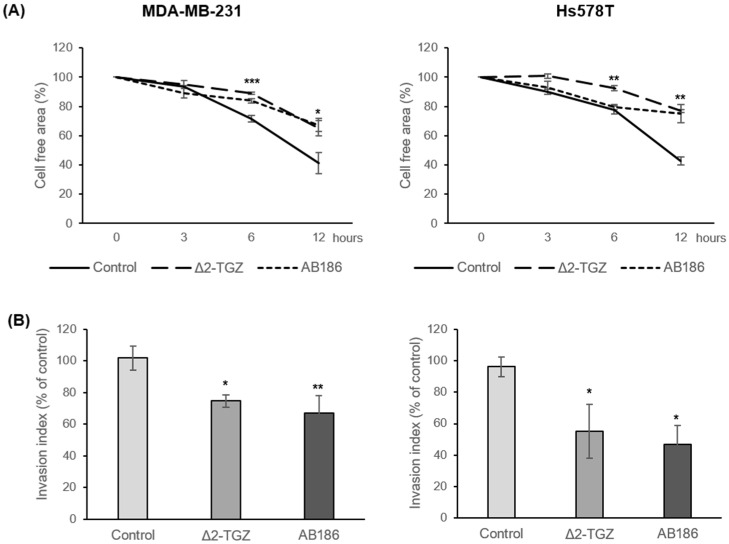
TGZ derivatives alter migration properties of MDA-MB-231 and Hs578T cells. MDA-MB-231 and Hs578T cells were exposed to DMSO as control, or 20 µM of Δ2-TGZ or 4 µM of AB186 and subjected to wound healing (**A**) or transwell migration (**B**) assays. (**A**) Cell free area was measured after 3 h, 6 h, and 12 h of TGZ derivatives exposure. Each wound healing corresponds to the mean of three different areas. (**B**) Cells were seeded in the upper chamber of Transwell membranes and treated with Δ2-TGZ or AB186 for 9 h. Graphs represent the average of the migrated cells number in the lower chamber. Reported data are the means of three or six experiments. Analysis of variance (ANOVA) with Bonferroni post-hoc test was used to determine significant difference of migration from control cells, where * *p* < 0.05, ** *p* < 0.01, and *** *p* < 0.001.

**Figure 7 ijms-23-06859-f007:**
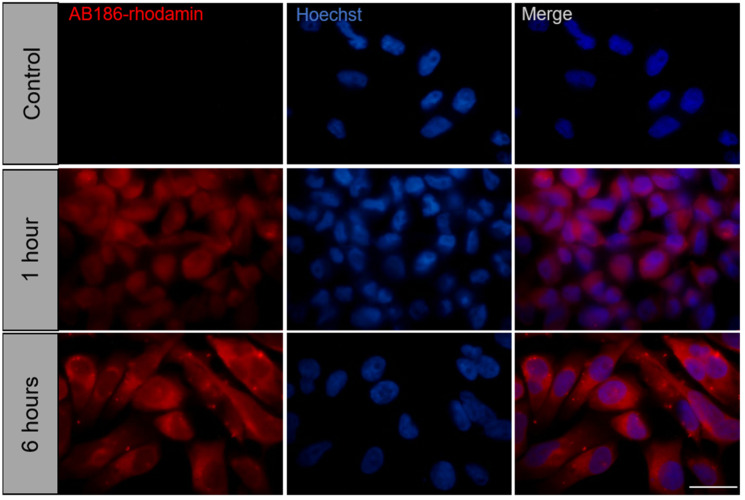
Rhodamin-labelled AB186 localizes in the cytoplasm of MDA-MB-231 cells. MDA-MB-231 cells were exposed to 3 µM of Rhodamin-labelled AB186 during 1 or 6 h. Control cells were exposed to the unlabeled compound. Cells were fixed with paraformaldehyde and cell nuclei were labelled with Hoechst dye. Bar represents 50 µm.

**Figure 8 ijms-23-06859-f008:**
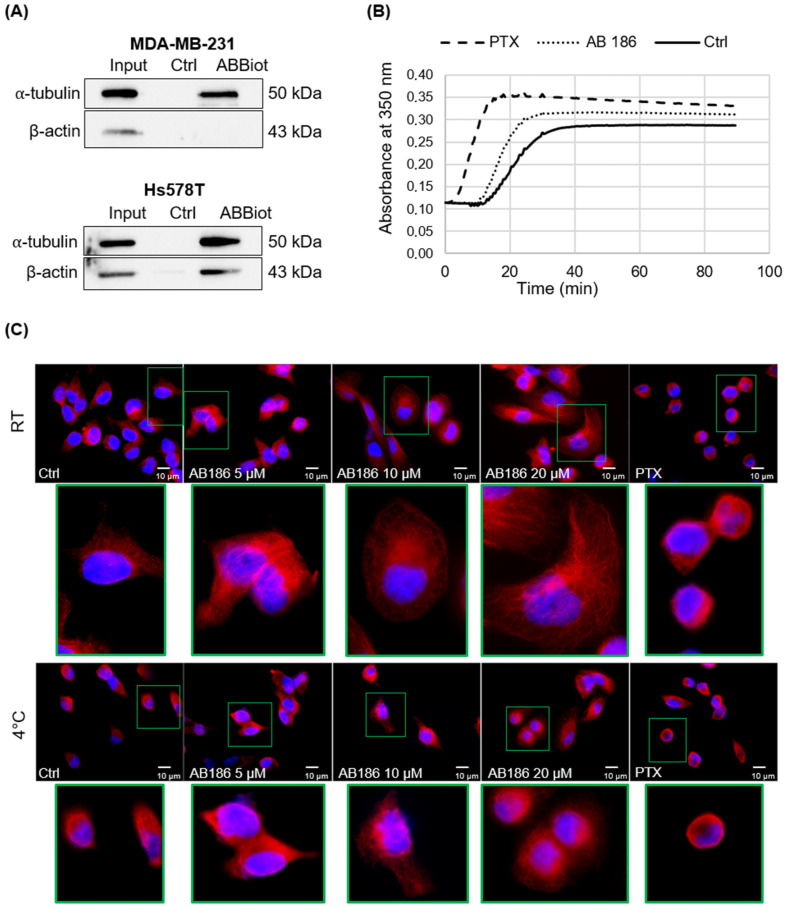
AB186 directly targets the microtubule cytoskeleton. (**A**) MDA-MB-231 and Hs578T cells lysate were incubated with the microbeads and biotinylated AB186 (ABBiot) for 40 min or in absence of AB186 (Ctrl). Precipitated proteins were then subjected to western blot analysis. Western blot analysis was performed using anti-α-tubulin and anti-β-actin antibodies. Input corresponds to the total lysate before pull-down assay. (**B**) In vitro tubulin polymerization assay was performed. β-tubulin was exposed to DMSO (Ctrl), AB186 (5 µM), or paclitaxel (10 µM). GTP was added to initiate the reaction. The tubulin polymerization rate was monitored for 90 min at 37 °C and measured by absorbance at 350 nm. (**C**) MDA-MB-231 cells were treated with DMSO (Ctrl), AB186 (5, 10 or 20 µM), or paclitaxel (PTX) (5 µM) for 15 min at room temperature (RT) or 4 °C. Then, cells were subjected to immunofluorescence analysis after staining with antibodies raised against α-tubulin. Cell nuclei were labelled with Hoechst dye. Representative high-resolution image focusing on one single cell in each group present a 6X magnification. Bar represents 10 µm.

**Table 1 ijms-23-06859-t001:** Maximal difference of cell index between control condition and treated condition, Δ2-TGZ/Control and AB186/Control, was determined for three independent experiments. The mean of maximal difference of CI for each treatment was determined in MCF-7 and MDA-MB-231 cells.

		Time of CI Maximal Difference		Time of CI below Control	
Cell Line	Experiment	∆2-TGZ/Control	AB186/Control	∆2-TGZ	AB186
MCF-7	1	2 h	2 h 45 min	6 h	4 h 30 min
	2	1 h 45 min	4 h	3 h 30 min	6 h 45 min
	3	2h	5 h 15 min	3 h 45 min	9 h 30 min
	Mean ± SD	1 h 55 min ± 8 min	4 h ± 75 min	4 h 25 min ± 82 min	6 h 55 min ± 150 min
MDA-MB-231	1	1 h 45 min	3 h 30 min	5 h	7 h 15 min
	2	1 h 45 min	4 h	3 h 45 min	7 h
	3	2 h	4 h 15 min	5 h 45 min	8 h 15 min
	Mean ± SD	1 h 50 min ± 5min	3 h 55 min ± 13 min	4 h 50 min ± 60 min	7 h 30 min ± 40 min

**Table 2 ijms-23-06859-t002:** Cystoskeleton proteins identified by MS analysis using the Proteome Discoverer software. Corresponding peptides were obtained after biochemical treatments of the captured cytoplasmic proteins from MDA-MB-231 breast cancer cells on AB186-hybridized SPR chip. # Peptides: number of different peptides corresponding to a unique protein. Mascot Score: statistical score based on the probability that the peptides from the sample match those in the selected protein database.

Protein		UniProt ID	# Peptides	Mascot Score
Actin	ACTG1	P63261	8	316.4
Tubulin beta chain	TUBB	P07437	8	144.9
Filamin-A	FLNA	P21333	7	82.8
Tubulin alpha chain	TUBA1C	Q9BQE3	4	75.6
Moesin	MSN	P26038	4	65.9
Talin-1	TLN1	Q9Y490	8	55.3

## Data Availability

Not applicable.

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
