# Peer review of "AB186 Inhibits Migration of Triple-Negative Breast Cancer Cells and Interacts with α-Tubulin"

_ijms, 2022, doi:10.3390/ijms23126859_

Round 1

Reviewer 1 Report

In this manuscript entitled “AB186 inhibits migration of triple-negative breast cancer cells and interacts with α-tubulin”, Geoffroy et al. has investigated the antitumor effect and potential mechanistic actions of two Troglitazone derivatives Δ2-TGZ and AB186. They first showed both chemicals could induce apoptosis in breast cancer cell line MDA-MB-231 but not in normal breast cells. Then they observed rapid change in cellular impedance upon drug treatment, and reasoned that both chemicals could affect cell morphology, which was proved by using immunofluorescence staining. They have also discovered that both chemicals AB186 could inhibit the migration of two triple negative breast cancer cell lines. To further investigate the mechanistic action of the AB186, they performed SPR spectrometry and identified the interaction between AB186 and alpha tubulin, and later found AB186 could affect the dynamics of microtubule network by enhancing tubulin polymerization.

In general, the experiments of this study are well designed, the data is of good publishable quality and well presented, and the manuscript is written and formatted clearly and precisely. I recommend accepting this manuscript for the publication in IJMS if the authors can address discuss my questions bellow:  

1.     The usage of cell lines was not consistent in these functional experiments of this study. MCF-7 was used for RTCA and morphological analysis, MDA-MB-231 was used for all experiments, and Hs578T was used for migration, SPR spectrometry and IP validation. Could the authors explain why you didn’t keep using same cell lines through this study even if your main observations (apoptosis and RTCA) may look like universal among different breast cancer cell lines? Especially the reason for changing from MCF-7 to Hs578T for migration after getting RTCA data with MCF-7? It is feasible to perform wound healing with MCF-7 as I know.

2.     The authors showed solid results on the interaction between AB186 and alpha tubulin and the stabilization of microtubule with the drug. I just wonder if you by any chance have checked if AB186 could affect the progression of mitosis (metaphase) as other microtubule stabilizing reagents do such as paclitaxel. Do you think this might be one of the actions of this chemical for inducing apoptosis?

3.     In Figure 8C, the authors performed immuno staining to check the impact of AB186 on the microtubule network. Could you explain why this experiment was done at both room and low temperatures, and maybe update the manuscript with the explanation if you think it’s necessary?

Reviewer 2 Report

Dear Authors,

I congratulate you on the interesting topic. The paper is an in vitro study to better understand the mechanism of action of two TGZ derivatives (Δ2-TGZ and AB186) on the inhibition of triple negative breast cancer (TNBC) proliferation. In particular, your data showed that AB186 was a potent pro-apoptotic agent by interacting on α-tubulin, even if the exact consequences of this interaction needs to be further evaluated.

The paper is well written, original and within the scope of the journal. It provides relevant data that could be a stimulus towards a more in-depth evaluation with an in vivo study.

Please, avoid bold typing in the abstract and correct font dimension on lines 218-219.

Given these considerations I think that your paper could be accepted for publication in present form.

Kind Regards

Round 2

Reviewer 1 Report

The authors have addressed my questions well and updated the manuscript accordingly. I suggest to accept this manuscript in current form.